# Development of Bidirectional Pulsed Power Supply and Its Effect on Copper Plating Effect of Printed Circuit Board Via- Filling

**Wenguang Chen** * , **Shoutao Wang** , **Zhijian Liu** , **Caiyi Wei and Yuanyuan Peng**

School of Electrical Engineering, University of South China, Hengyang 421001, China
* Correspondence: chenwg@usc.edu.cn; Tel.: +86 7348282775

**Abstract:** A bidirectional pulse power supply with continuously adjustable forward parameters 8 V/20 A and reverse parameters 20 V/50 A was designed using DSP (Digital Signal Processor), and the bidirectional pulse power supply was used to test copper plating on printed circuit boards with filled via holes. The effects of frequencies, pulse width ratios of forward and reverse currents, and current densities on the copper plating effect were investigated by the single variable method and were compared with DC copper plating. The experimental results showed that compared with DC power supply, the bidirectional pulse power supply had a better effect and a faster speed on via-filling copper plating, and can also reduce the use of additives, which is in line with green development. The parameters of the pulse affected the plating effect to varying degrees. In this solution system, the optimal parameters for bidirectional pulse plating are frequency 1 kHz, forward pulse current density 4 ASD (Ampere per Square Decimeter) with 50% duty cycle, and reverse pulse current density 16 ASD with 2.5% duty cycle.

**Keywords:** bidirectional asymmetric pulse power supply; printed circuit board; via-filling copper plating; LLC resonant converter





## 1. Introduction

In the 1940s, Dr. Panl Eisler, the "father of printed circuit" in Britain, first proposed the concept of printed circuit boards (PCBs) and applied it to radios successfully. After research and decades of production practice, the printed circuit industry has achieved tremendous development. Electronic products are becoming lighter, thinner, and smaller, while their performance is developing towards multi-functional, high density, and high speed. This development trend requires continuously decreasing the size of the printed circuit board, improving the density of printed circuit board wiring, and reducing the size of the board micro-hole (buried, blind, via holes) to become an effective means to achieve high-density interconnection [1].

In the past, most PCB holes were filled with resin and conductive adhesive, but these had the disadvantages of high cost, complex processing, poor reliability, and environmental pollution. Later, the direct electroplating copper filling technology emerged, which has the characteristics of high reliability, good thermal conductivity, and low cost [2–6]. The reduction of the hole diameter makes the hole thickness-to-diameter ratio (AR) larger and larger (generally, blind holes plating with AR above 0.7 is called high thickness-to-diameter plating), which makes it more and more difficult to seal the hole with copper [7]. In the micro-via area, the non-uniformity of the current density distribution causes the deposition rate of copper on the surface to be greater than the deposition rate of the hole wall and bottom. Therefore, it is extremely easy to form holes or seams during deposition, and this result adds a lot of unreliability factors to the use of printed circuit boards [8–13]. Currently, a DC power supply or pulse power supply is generally used for hole filling and copper plating. A DC power supply has the advantage of simple structure and low cost, but plating

with DC takes a long time and requires a certain amount of additives to solve the problem of uneven coating and seams. Adding additives will improve the deposition effect, so that the deposition rate at the bottom of the micro-hole is higher than the deposition rate on the surface to finally obtain a more ideal filling effect. However, excessive use of additives will pollute the environment, and, even if additives are used, there is still an upper limit on the efficiency and quality of hole-filling copper plating using DC power [14–18]. Using a bidirectional pulse power supply for hole-filling copper plating, the cost is high and the production is complicated, but it has the advantages of uniform and detailed coating, high deposition efficiency and low porosity, and it only needs a small amount of additives or no additives at all, which is in line with the concept of green development [19–21]. Therefore, it is necessary to study the effect of a bidirectional pulse power supply on hole-filling copper plating, to further improve the efficiency and quality of hole-filling copper plating and promote green development.

Low-voltage high-current pulse power supplies generally uses the phase shift full-bridge topology or LLC topology. Phase shift full-bridge topology has a wide output range and high efficiency, which can meet customer needs to the greatest extent, so it is used widely [22–25]. The phase-shifting full bridge can only achieve zero voltage turn-on, while the LLC can achieve zero voltage turn-on and zero current turn off at the same time, so the efficiency of the LLC is relatively higher, but the output range of the LLC is small relatively, and the efficiency will be very low under light loads [26]. As this type of pulse power output is low voltage, high current, when the current is large enough, the method of staggered parallel connection is used to increase the output current and reduce the output current ripple. The interleaving technique doubles the output current at the equivalent frequency, reduces the output current ripple, and optimizes the thermal effect and power distribution [27–30]. However, the tolerance of the resonant slot leads to different conversion gains and unbalanced output loads in each phase. The imbalance of the output current increases the output current ripple, making one of the phases transmit most of the load current, which may lead to equipment failure. Therefore, the current sharing mode must be adopted when paralleling [31–35].

In this paper, a bidirectional pulse power supply was designed and made, and then a bidirectional pulse plating technique was used to plate an acidic copper sulfate solution. By changing the frequency, pulse current, pulse width ratio, and amplitude ratio, and comparing with DC electroplating, the effect of bidirectional pulse power supply on via-filling copper plating was systematically investigated.

## 2. Principle of Bidirectional Pulse Plating

While DC electroplating can only adjust voltage or current, bidirectional pulse plating has independent parameters such as forward and reverse pulse current density, frequency, and pulse width ratio. The current density is the ratio of the current amplitude to the plated area. What affects the copper plating rate is the current density, not the current amplitude.

A schematic diagram of the bidirectional pulse via-filling copper plating process is shown in Figure 1. The left part is a bidirectional pulse waveform, the right part is a cross-section diagram of via holes in a printed circuit board, and the yellow part represents the plated copper. During the forward pulse conduction, the printed circuit board acts as the cathode and the copper starts to be deposited on its surface. The pulse current density can be several times or even ten times that of the DC density. The high current density makes the deposition rate of copper faster and forms a deposition layer with a finer grain structure. However, the high current density causes the metal ions near the cathode to be consumed at a very fast rate, and the concentration of metal ions near the cathode drops to a very low level, and the potential decreases from outside the via holes to inside, so the thickness of the copper wall from inside the via holes to outside also increases correspondingly. When the pulse enters the off period, due to the concentration difference, the metal ions will diffuse to the cathode, so that the concentration of metal ions near the cathode will rise, accompanied by phenomena such as recrystallization, adsorption, and desorption, which

are beneficial to the coating [14]. The short reverse pulse has a large amplitude, and during the reverse pulse conduction, the printed circuit board acts as the anode, and the anode current distribution is highly uneven, which dissolves the coating projections and makes the surface of the coating smoother and flatter. In addition, the interchange of polarities also makes the dispersion ability of the plating solution stronger [15].

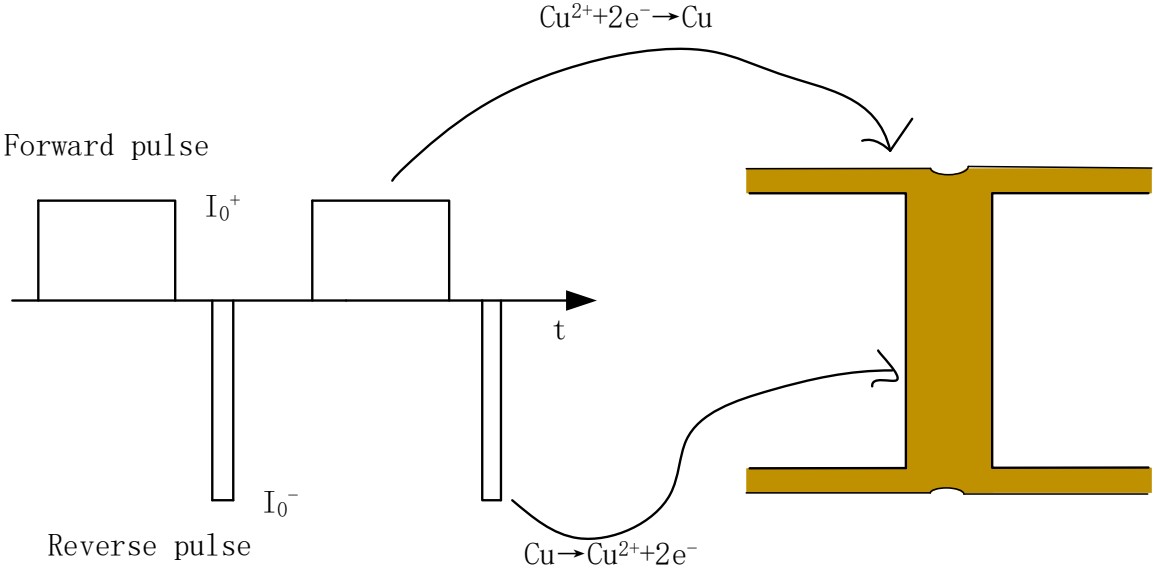

**Figure 1.** Principle of bidirectional pulse via-filling copper plating process.

Compared with DC plating, bidirectional pulse plating has the advantages of fast coating deposition rate, fine coating crystallization, strong dispersion ability of the plating solution, high coating purity, and low hydrogen embrittlement of the plating layer [16,17]. Therefore, the efficiency of bidirectional pulse copper plating will be higher, and when the coating effect is similar, the use of additives can be greatly reduced, in line with the concept of green development.

### 3. Development of Bidirectional Asymmetric Pulse Plating Power Supply

The basic structure diagram of the bidirectional pulse plating power supply is shown in Figure 2. The circuit mainly includes two DC power supplies, a pulse generation circuit, an auxiliary power supply, a DSP controller, a sampling circuit, and a driving circuit. In order to reduce the switching loss and improve the reliability of the DC power supply, the DC component after the rectification of the power grid passes through two LLC resonant converters to generate two low-voltage and high-current outputs that do not interfere with each other. The output of the forward LLC resonant circuit is connected to the bridge arm of Q9 and Q12, which outputs a forward pulse of 8 V/20 A to the load; the output of the reverse LLC resonant circuit is connected to the bridge arm of Q10 and Q11, which outputs a reverse pulse of 20 V/50 A to the load. In this way, the bridge is used to generate bidirectional pulses during the pulsed via-filling copper plating process to isolate the forward and reverse pulses and ensure that the copper deposition and dissolution processes do not interfere with each other. The main controller uses DSP28335, which can output 12 PWM waves, of which 6 can achieve complementary output. It is very suitable for this kind of application where multiple PWM waves are needed. It is simple, stable, and reliable to control.

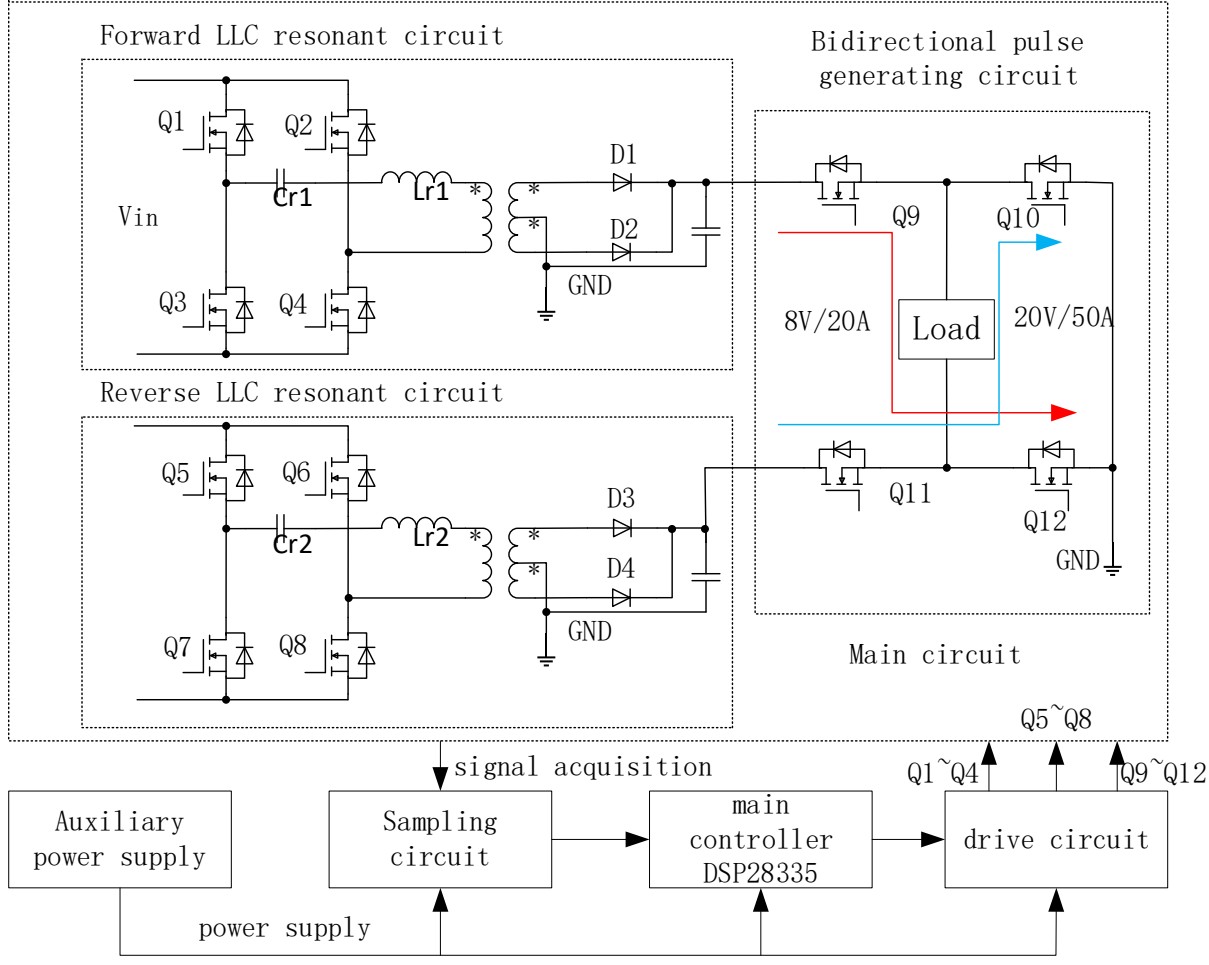

**Figure 2.** Bidirectional pulse power system structure diagram.

### 3.1. LLC Resonant Converter Design

According to the demand for copper plating of printed circuit board filling, the LLC resonant converter structure is used. LLC has the characteristics of zero voltage turn-on and zero current turn-off, high efficiency [18–20], and the efficiency of the power supply system is not less than 90%. The grid voltage rectified and filtered is 310 ± 31 V, the power conversion efficiency is taken as 92%, the maximum output of the forward converter is 8 V/20 A, the maximum input power of the forward LLC resonant converter is calculated as 174 W, and the design of the switch uses the STP18N65M2 type MOSFET with a maximum current of 12 A and a breakdown voltage of 650 V; the maximum output of the reverse converter is 20 V/50 A, the maximum input power of the reverse LLC resonant converter is 1087 W, and the design of the switch uses a SIHA25N60EFL type MOSFET with a maximum current of 25 A and a breakdown voltage of 650 V. The minimum voltage gain at the maximum input voltage is determined to be:

$$M_{min} = \frac{k+1}{k} \tag{1}$$

The maximum gain is:

$$M_{max} = \frac{V_{inmax}}{V_{inmin}} M_{min} \tag{2}$$

where $V_{inmax}$ is the maximum input voltage value; $V_{inmin}$ is the minimum input voltage value; and $k$ is the ratio between the excitation inductance and the primary magnetic leakage inductance, generally taking 5~10, where $k$ is taken as 7, the minimum gain is 1.14,

and the maximum gain is 1.36. Considering a 10% margin, according to the peak gain curve, Q is taken as 0.3, and Q is the quality factor. The transformer turn ratio is determined according to the following formula:

$$n = \frac{V_{inmax}}{V_o + V_F} \cdot M_{min} \tag{3}$$

where $V_F$ is the voltage drop of the secondary rectifier diode and $V_o$ is the output voltage. The turn ratio of the forward LLC converter transformer is calculated to be 41, and the turn ratio of the reverse transformer is 18. The Equivalent Load Impedance is calculated using Fundamental Wave Analysis Method:

$$R_{\text{ac}} = \frac{8n^2 V_o{}^2}{\pi^2 P_o} \eta \tag{4}$$

where $P_o$ is the output power and $\eta$ is the power conversion efficiency. It is calculated that the equivalent load impedance of the forward LLC converter is 519 Ω and that of the reverse LLC converter is 99 Ω. Then, calculate the parameters of the resonant circuit according to the following formulas:

$$C_r = \frac{1}{2\pi Q f_0 R_{ac}} \tag{5}$$

$$L_r = \frac{1}{(2\pi f)^2 C_r} \tag{6}$$

$$L_p = \frac{(k+1)^2}{(2k+1)L_r} \tag{7}$$

Assuming the resonant frequency $f_0$ = 100 kHz, we can calculate $C_r$, $L_r$, and $L_p$. In order to ensure that the resonant point of the light load and heavy load is at the set value, it is necessary to evaluate the resonant state of the converter under different load conditions, and account for $C_r$, $L_r$, and $L_p$ $L_p$ according to the load gain curve. If the calculated resonant point is not at 100 kHz, it is necessary to go through several iterations again to ensure that the resonant parameters of the forward LLC converter are 10 *nF*, 250 *μ*H, and 116 *μ*H, and the resonant parameters of the reverse LLC converter are 64 *nF*, 40 *μ*H, and 208 *μ*H, respectively. The resonant waveform of the actual test is shown in Figure 3, and the resonant current $I_r$ lags behind $V_{ds}$ (voltage at both ends of MOS tube drain and source) to achieve zero voltage turn-on. Since copper plating requires a wide output range and the efficiency of the LLC converter is very low at light loads, the control strategy of "hiccup" mode is used to reduce the driving duty cycle of the LLC converter at light loads to ensure that the system can operate efficiently at light loads.

### 3.2. Bidirectional Pulse Generator Design

The pulse current is formed by chopping the DC through the power switches. The different low-voltage high currents generated by the two LLC converters are applied to the two bridge arms of the four-switch bridge circuit, and the output of the bidirectional pulse current is controlled by the DSP, which generates repeat frequency, duty cycle, and sequential of the bridge arm drive waveform. As shown in Figure 4, a bidirectional pulse current waveform with a forward current of 15A and a reverse current of 37 A is the output. The efficiency of the overall system can reach up to 93%.

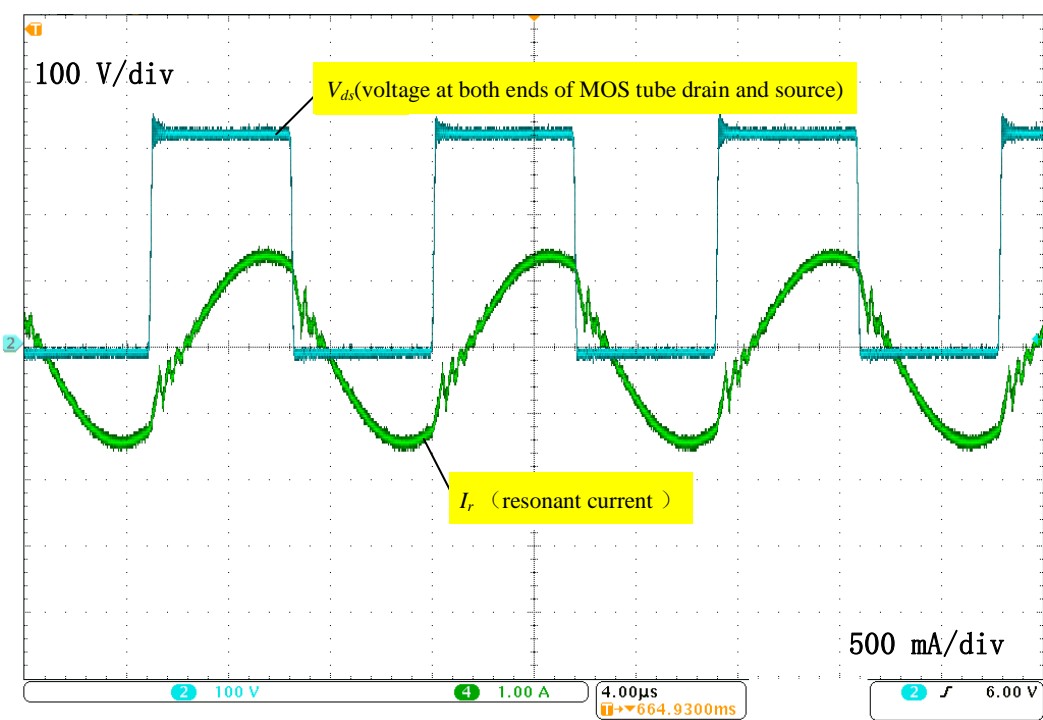

**Figure 3.** Zero voltage turn-on waveform.

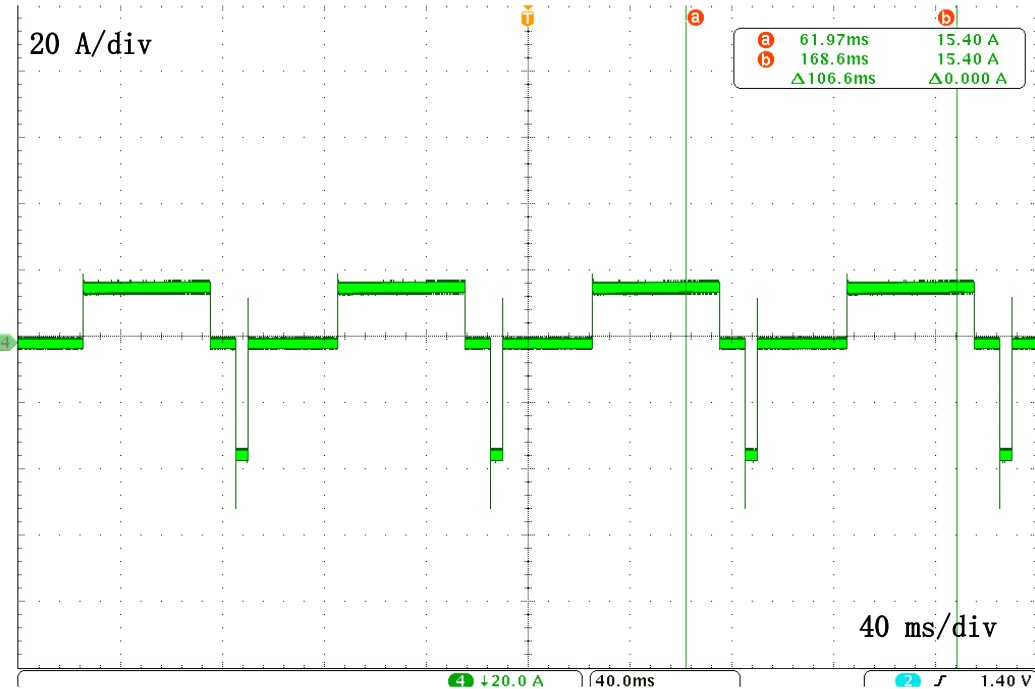

**Figure 4.** Bidirectional asymmetric pulse current waveform.

### 3.3. Pulse Current Closed-Loop Control Strategy

The decisive parameter in the plating effect is the current density. To get the ideal plating effect, the current amplitude must be controllable and constant, so it is necessary to do pulse current closed-loop control, and a high-quality pulse current closed-loop control strategy corresponding to the pulse power supply for pulse hole-filling copper plating becomes crucial. While DC current closed-loop control is sampled and adjusted in real

time, pulse current closed-loop control cannot do so, because in this case, during the pulse off interval the sampled current value is 0, and the closed-loop operation will adjust the output voltage to very high level Then, when the next pulse comes, the voltage will return from a maximum value to the normal value, which will lead to the pulse with a particularly spike. Furthermore, we cannot use the so-called current average method, the average of the sampling value of several cycles, and then use this average to do closed-loop regulation. Although the pulse and the turn off period are of the same frequency in the same cycle, the turn off period is equivalent to no load. When the drive frequency remains unchanged, the output voltage of the LLC will rise when it is no load, so the output current will still have pulse spikes, which are much smaller than the pulse spikes of the previous method. Because the average value of several cycles is used for closed-loop control, this control strategy will make the amplitude of pulse current in a single cycle unstable. In order to ensure constant pulse current amplitude and no pulse spikes, an intelligent PI closed-loop control strategy is proposed and its flow chart is shown in Figure 5 below. First, a PWM (Pulse Width Modulation) wave at a frequency of $f_0$ is provided, and then the output current $I_0$ is detected. When $I_0$ is not equal to 0, the normal PI (Proportional Integral) adjustment is made. When $I_0$ is equal to 0, the PWM wave and PI operation are turned off and the I value is retained. This is different from the DC current closed-loop control, and the most critical point is to turn off the PWM wave during the pulse turn off, so that the output of the LLC is 0, and the pulse spike will not occur at the beginning of the next pulse. In addition, the PI operation is turned off, and the I value is retained, so that the next pulse can continue to adjust or stabilize the value of the previous pulse.

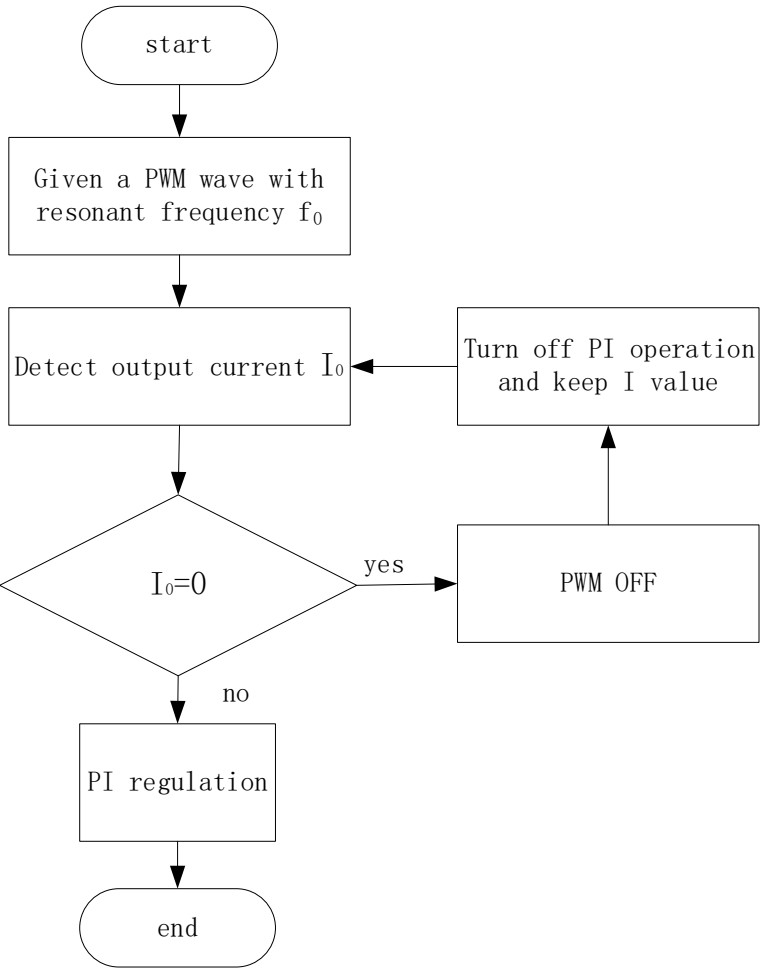

**Figure 5.** Flow chart of intelligent PI closed-loop control strategy.

Figure 6 is the waveform diagram of pulse current intelligent PI closed-loop control under digital closed-loop simulation. The first waveform is the voltage and current waveform of the LLC resonant cavity. There is no voltage and current in the resonant cavity during the pulse off period, indicating that the PWM is turned off; the second waveform is the output pulse current waveform. Load interference is added at 0.041s in the model. It can be seen that the output current returns to the set 20 A within 10 ms, which is very stable without a pulse spike; the third waveform is the count value corresponding to the drive frequency.

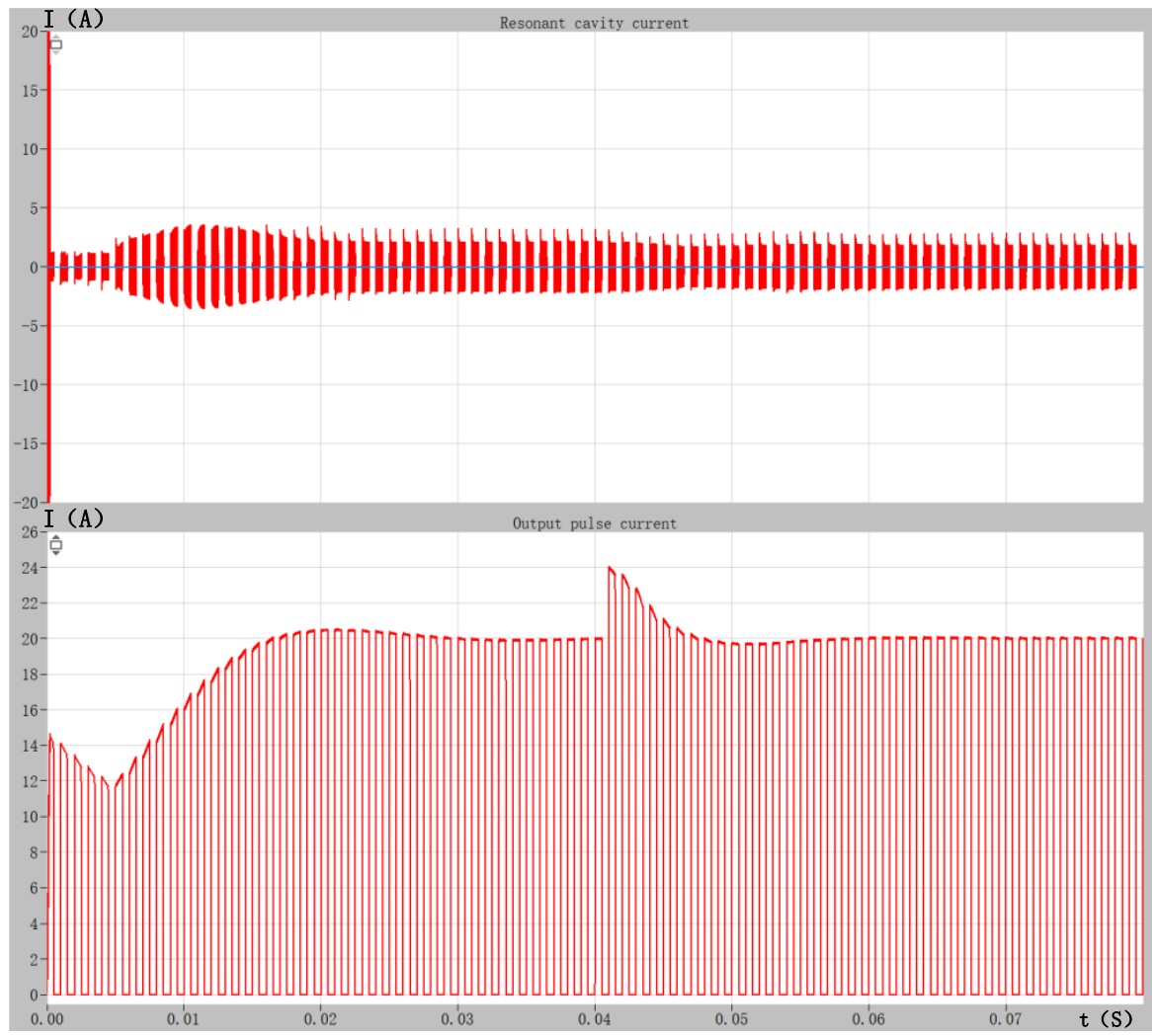

**Figure 6.** Digital simulation waveform of intelligent PI closed-loop control strategy. (Top wave is the resonant cavity current, bottom wave is the output current).

## 4. Plating Test Experiment

The electroplating tank model is shown in Figure 7. A copper board containing phosphorus was used as the soluble anode with a size of 50 mm × 70 mm × 0.3 mm, and the cathode was a printed circuit board of 35 mm × 70 mm × 1.6 mm. Each printed circuit board was evenly distributed with an array of 12 × 32 via holes; the via holes diameter is 0.4 mm, depth is 1.6 mm, and spacing is 2 mm. The electrolytic tank is a 120 mm × 100 mm × 100 mm quartz tank; the anode was hung on both sides, the cathode was fixed in the middle, and the distance between the two stages was 60 mm. The plating solution was proportioned to 80 g/L, 210 g/L, and 60 mg/L. The developed bidirectional pulse power supply was applied to provide excitation for via-filling copper plating of the printed circuit boards. The hole-filling copper plating experiments were carried out

at room temperature, each plating time was controlled at 40 min, and 4 g of was added to the electrolytic tank after two plating tests. The electroplating parameters were set as: forward pulse frequency 1 kHz, duty cycle 50%, and current density 4 ASD; the reverse pulse frequency 1 kHz, duty cycle 2.5%, and current density 16 ASD. First, we compared the results of DC hole-filling copper plating with current densities of 4 ASD and 0.5 ASD. Then, we changed the frequency, forward and reverse pulse duty cycle ratio, and current density. The effect of frequency, duty cycle ratio of forward and reverse pulses, and current density on the effect of hole-filling copper plating was investigated by the single variable method.

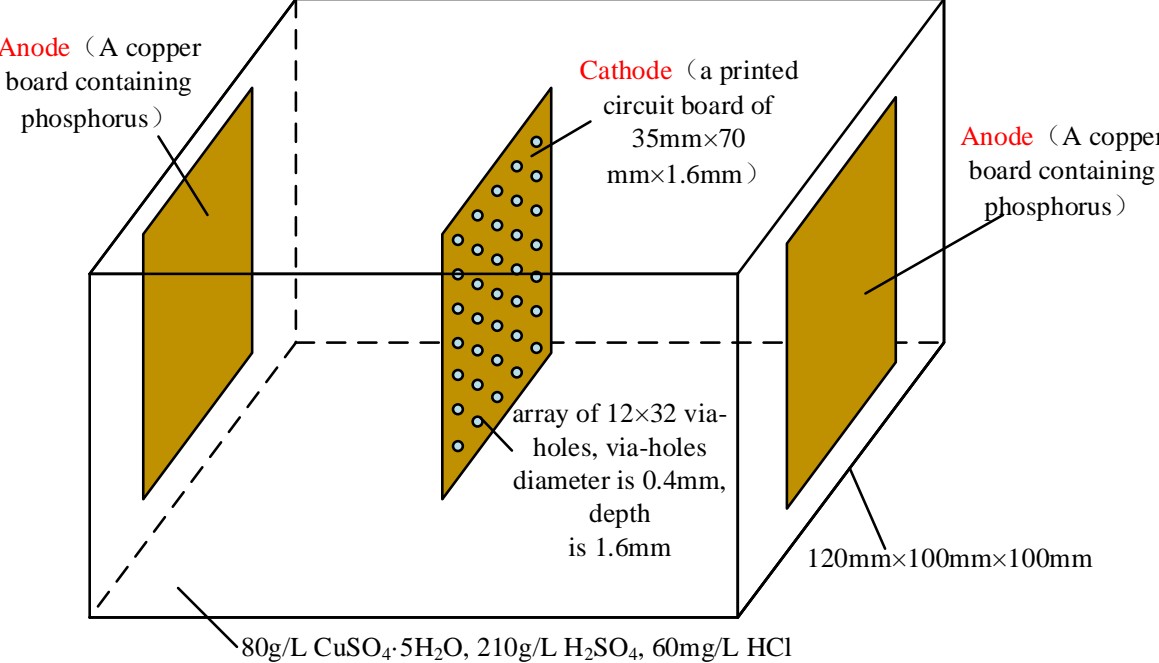

**Figure 7.** Electroplating tank model diagram.

*4.1. Comparison of The Effects of Pulse Plating and DC Plating on Via-Filling Copper Plating*

The samples were scanned using a Quanta 250 scanning electron microscope (SEM). Figure 8a–c sequentially shows the bidirectional pulse power supply (frequency of 1 kHz, forward pulse duty cycle of 50%, current density of 4 ASD, reverse pulse duty cycle of 2.5%, current density of 16 ASD) and the DC power supply of the current density of 4 ASD and 0.5 ASD. After 40 min of electroplating, the coating was magnified by 1000 times, and the surface morphology scale is 100 μm. On the flat surface after pulse plating, the surface particles were small and tightly packed, and there were few voids; the surface of the plating by DC plating with 4 ASD current density was rough, the surface particles varied in size with the large ones reaching 10 μm, and the loosely packed voids were large. The surface of the plating by DC plating with 0.5 ASD current density was flat, there were a few depressions, and the surface particles were very small and tightly packed. For DC plating with 4 ASD, the current density was too high, the deposition rate was too fast, and the current flow was continuous unidirectional, so the plating solution was poorly dispersed, resulting in large surface particles and a loose accumulation of electrodepositing. The DC plating with 0.5 ASD has a low current density and slow deposition rate, so the surface accumulation was tight and had small voids. However, compared with pulse plating, the plating thickness was less than one quarter that of pulse plating.

Since the selected via holes size is large, with a via hole diameter of 400 μm, and the plating thickness was between 20 μm and 30 μm, it was difficult to see changes in plating thickness inside the via holes in the cross-sectional view of the whole via holes, so a section of the via hole orifice was cross-sectioned to observe the change of plating thickness from outside to inside the via holes. Figure 9a,b shows the cross-sectional view of the via-holes orifice at a frequency of 1kHz, forward pulse duty cycle 50%, current density 4 ASD, reverse pulse duty cycle 2.5%, current density 16 ASD in both directions, and DC plating at a current density 4 ASD for 40 min at a magnification of 500 times and the surface morphology scale is 300 μm. As shown in Figure 6a, the plating thickness of the via holes from outside to inside was uniform, all 25 μm, with dense plating inside and a flatter surface under the action of the reverse pulse. Figure 6b shows that the plating inside the via holes is thinner, the protrusions increased, the surface flatness decreased, and the thickness of the plating from the outside to the inside of the via holes gradually decreased from 30 μm to 15 μm, indicating that the deposition rate became slower. This is due to the uneven distribution of current density and plating solution concentration during the DC plating operation, which caused the thickness of the plating near the via holes greater than the thickness of the plating inside the via holes; this unevenness is prone to a series of problems such as seams, affecting the quality of printed circuit boards. The bidirectional pulse copper plating is a printed circuit board process that acts both as a "cathode" electrodeposition of copper (positive pulse) and as an "anode" dissolution of copper (reverse pulse). Positive plating will also dissolve when thicker copper walls are deposited in the reverse plating, while the bidirectional current flow also improves the dispersion ability of the plating solution, so the plating layer of the whole via holes is more uniform under the action of a bidirectional pulse.

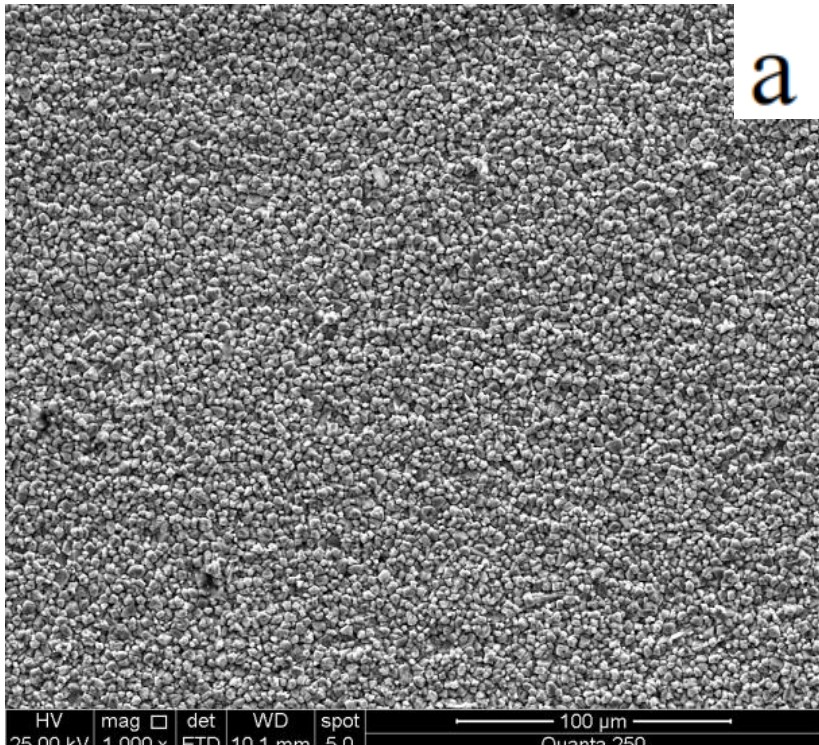

**Figure 8.** *Cont.*

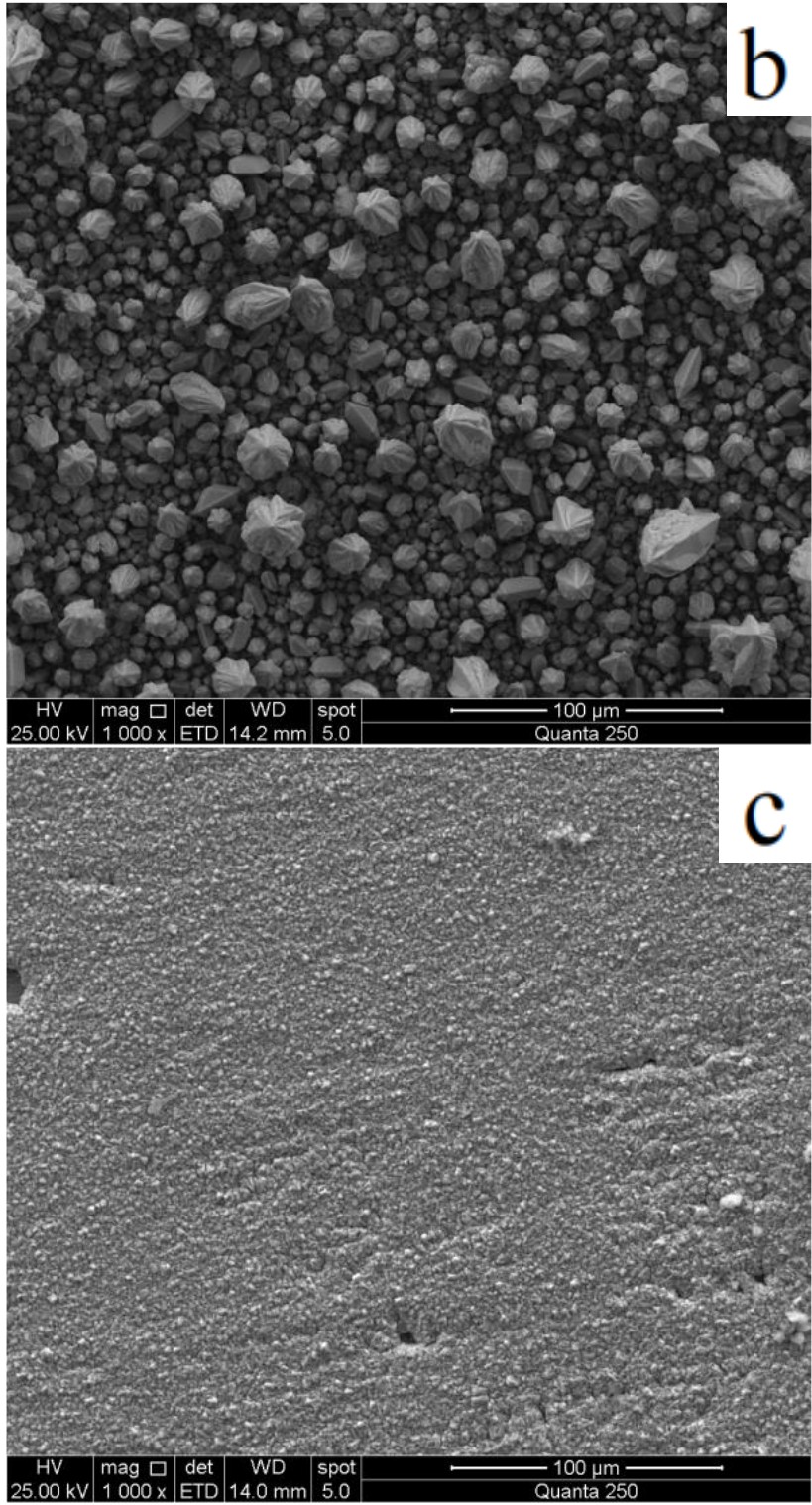

**Figure 8.** SEM scan comparison of plated surface under bidirectional pulse and DC electroplating.

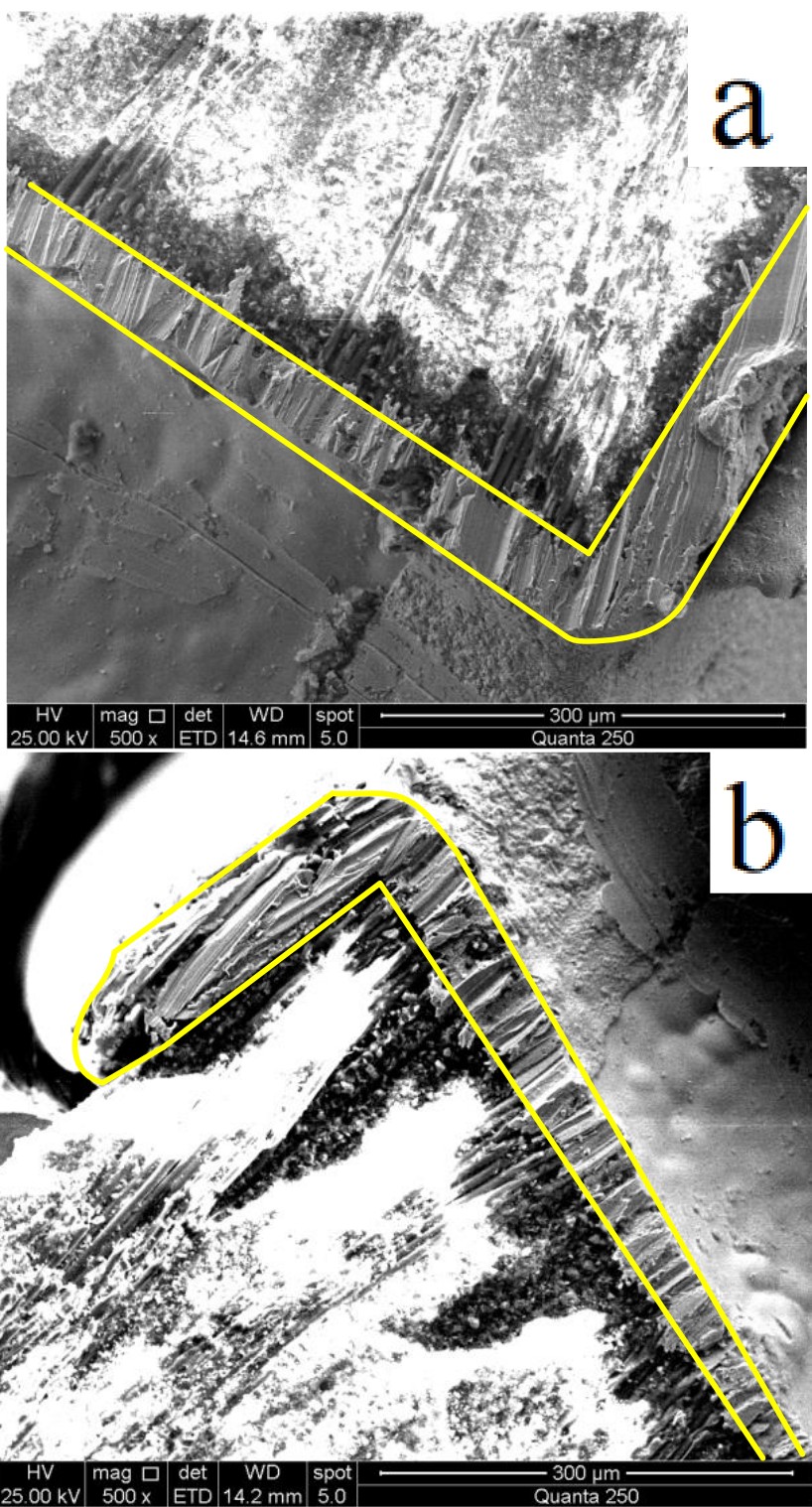

**Figure 9.** Cross-sectional view of via hole orifice under bidirectional pulse and DC plating.

*4.2. Effect of Pulse Plating Parameters on Via-Filling Copper Plating*

(1) To investigate the effect of frequency on via-filling copper plating, five samples were plated using a frequency of 100 Hz, 500 Hz, 1 kHz, 2.5 kHz, or 5 kHz, and plated for 40 min with a fixed forward pulse current density of 4 ASD and a duty cycle of 50% and a reverse pulse current density of 16 ASD and a duty cycle of 2.5%.

(2) To investigate the effect of forward and reverse pulse duty ratio on via-filling copper plating, five samples were plated for 40 min at a fixed frequency of 1 kHz with a forward pulse current density of 4 ASD and a duty cycle of 50% and a reverse pulse current density of 16 ASD. Three samples were plated with a forward and reverse pulse duty ratio of 50:1, 25:1, 20:1, 15:1, or 10:1.

(3) To investigate the effect of current density on via-filling copper plating, five samples with current densities of 1 ASD, 2 ASD, 4 ASD, 6 ASD, or 8 ASD were plated for 40 min at a fixed frequency of 1 kHz with a forward to reverse pulse amplitude ratio of 1:4, a forward pulse duty cycle of 50%, and a reverse pulse duty cycle of 2.5%.

The nine samples were scanned by an electron microscope to observe the size of the plated particles and the rate of change of the hole wall thickness; then, the effect of pulse plating parameters on via-filling copper plating was analyzed.

Figure 10 is a line graph of the effect of frequency, forward and reverse pulse duty cycle values, and current density on the size of the plated particle diameter. Figure 11 is a line graph of the effect of frequency, forward and reverse pulse duty cycle values, and current density on the rate of change of hole wall thickness. It can be seen from the graph that the effect of the electrical parameters on the size of the plated particle diameter was greater than the effect on the rate of change of the hole wall thickness, and the current density had a greater effect on the copper plating of the filled holes than the frequency and the value of the duty cycle of the forward and reverse pulses.

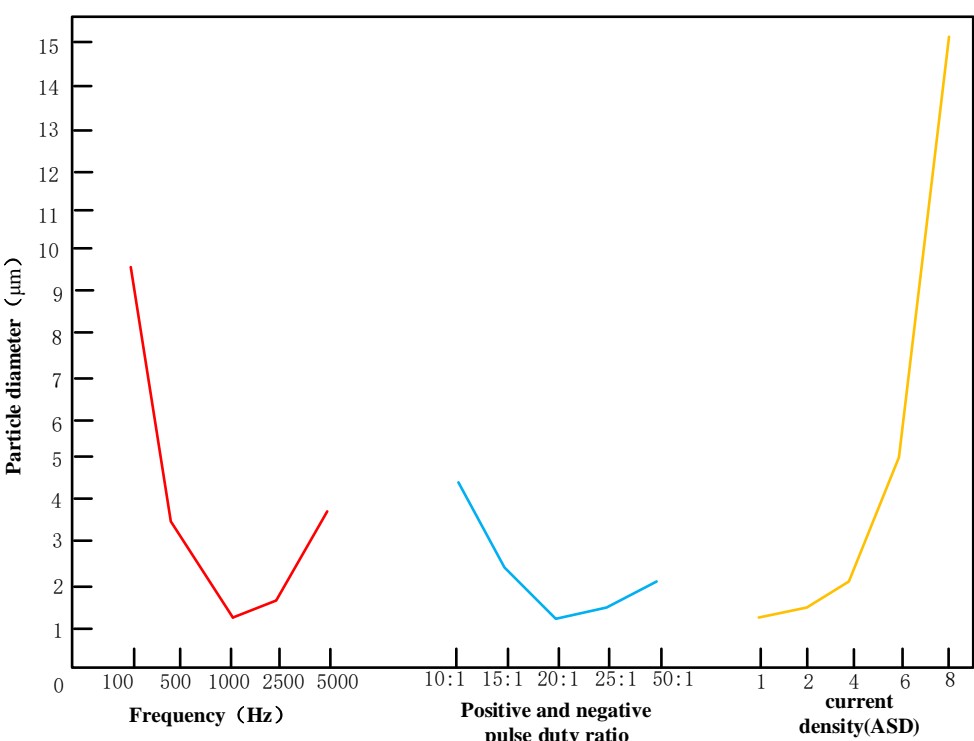

**Figure 10.** Relationship between pulse plating parameters and the rate of change of plating particle diameter.

In terms of pulse frequency, as the frequency increased, the size of the plated particle diameter and the rate of change of the hole wall thickness first decreased and then increased, which is due to the long single cycle time at low frequencies, which means that the single copper deposition time is also long, so the rate of change of the plating particles and the hole wall will be relatively large. However, when the frequency is too high, the single cycle time is short, the pulse off time and reverse time are short, and the ions in the plating solution cannot be dispersed in time; therefore, the dispersion ability of the plating solution is reduced, resulting in an increase of the plated particle diameter and the rate of change of

hole wall thickness. At 1 kHz, the size of the plated particle diameter was the smallest at 1.4 m, and the hole wall thickness variation rate was also the smallest.

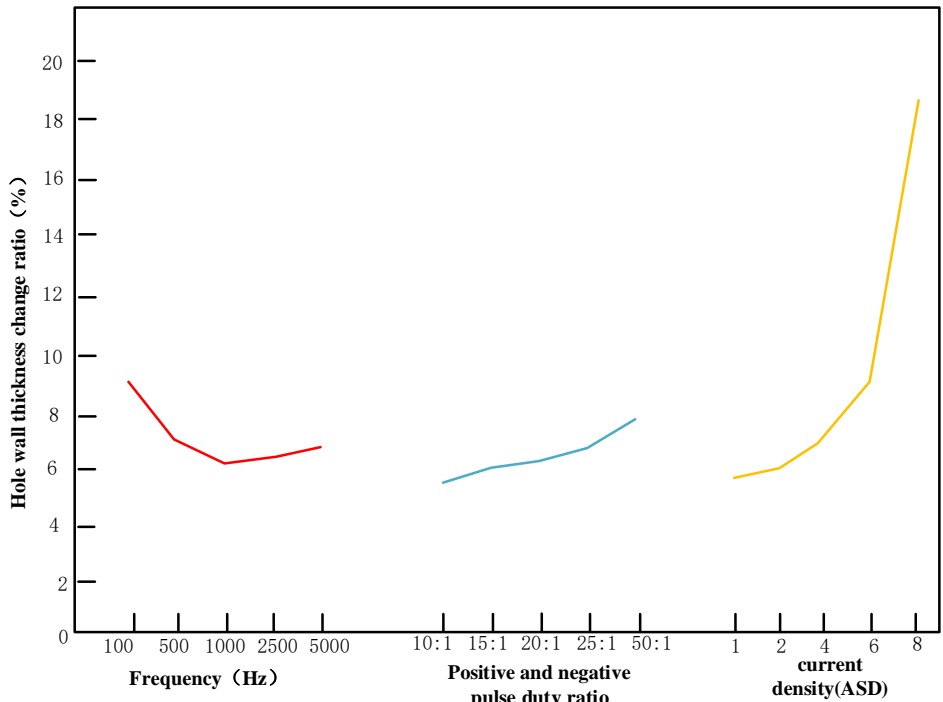

**Figure 11.** Relationship between pulse plating parameters and hole wall thickness.

As the duty cycle ratio of the forward and reverse pulses increased, the size of the plated particle diameter decreased first and then increased, and the rate of change of hole wall thickness increased continuously. The larger the ratio, the smaller the reverse pulse ratio, the smaller the reverse dissolution of copper, and the thickness of copper deposition was larger where the current density was high relatively, so the rate of change of the hole wall thickness was larger. When the ratio was 20:1, the particle diameter was the smallest, and when the ratio was 10:1, the change rate of the hole wall thickness was the smallest, but the particle diameter was too large. After comprehensive consideration, 20:1 was selected.

In terms of current density, with the increase in current density, the size of the plated particle diameter and the rate of change of the hole wall thickness increased. This is due to the large current density, the large polarization effect of the cathode, and the fast deposition speed. However, the current density on the cathode should not be too high and should not exceed the allowed upper limit. After exceeding the upper limit, due to the severe lack of metal ions near the cathode, a metallic layer shaped like a branch will be produced at the tip and protruding places of the cathode, or a loose layer shaped like a sponge will be produced on the whole cathode surface. When the current density was 1 ASD and 4 ASD, the effect of hole-filling copper plating was close, but the thickness of 1 ASD plating was only a quarter of that of 4 ASD, and 4 ASD was chosen.

## 5. Conclusions

A bidirectional pulse power supply with adjustable plating parameters was developed according to the copper plating process and its technical specifications were determined for filling holes in printed circuit boards. The device selection and parameter design process of the LLC circuit and pulse generation circuit were described in detail. The DSP was used as the controller to output a bidirectional pulse waveform with a periodic forward pulse of 8 V/20 A and a reverse pulse of 20 V/50 A. The developed bidirectional pulse power supply was used to conduct the via-filling copper plating experiments on printed circuit boards. The experimental results were compared with the effect of DC via-filling copper

plating, and then the effect of pulse electrical parameters on the electroplating was studied. The experimental results showed that the bidirectional pulse power supply was more effective and faster than the DC power supply for via-filling copper plating, which can reduce the use of additives and is in line with green development concepts. The electrical parameters affected the plating to different degrees. In this solution system, the optimal parameters for bidirectional pulse plating were frequency 1 kHz, forward pulse current density 4 ASD with 50% duty cycle, and reverse pulse current density 16 ASD with 2.5% duty cycle.

**Author Contributions:** Conceptualization, S.W. and W.C.; methodology, S.W.; software, Y.P. and S.W; validation, S.W., C.W. and Z.L.; formal analysis, W.C.; investigation, S.W.; resources, W.C.; data curation, S.W.; writing—original draft preparation, S.W.; writing—review and editing, Z.L. and C.W; visualization, S.W.; supervision, W.C.; project administration, S.W.; funding acquisition, S.W. and W.C. All authors have read and agreed to the published version of the manuscript.

**Funding:** This research was funded by the Hunan Graduate Scientific Research Innovation Project (223YSL009).

**Acknowledgments:** This work is supported Hunan Graduate Scientific Research Innovation Project.

**Conflicts of Interest:** The authors declare no conflict of interest.

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
