# Peer review of "Development of Bidirectional Pulsed Power Supply and Its Effect on Copper Plating Effect of Printed Circuit Board Via-Filling"

_electronics, doi:10.3390/electronics12030631_

Round 1
Reviewer 1 Report
Hello - I found your paper "Development of Bidirectional Pulsed Power Supply and Its Effect on Copper Plating Effect of Printed Circuit Board Via- Filling" interesting and informative. However, I've recommended revisions before publishing. There are many English grammar errors. Among these are:
“such as forward and reverses pulse current density,” should be “such as forward and reverse pulse current density.”
“Determine the minimum voltage gain at the maximum input voltage as…” where I think you really mean “The minimum voltage gain at the maximum input voltage is determined to be…”
Change “The decisive role in the plating effect is the current density.” to either “The decisive parameter in the plating effect is the current density.” or “The parameter that plays the decisive role in the plating effect is the current density.”
“current size” —> “current amplitude”
“And cannot use the…” —> “Furthermore, we cannot use the…”
“First, give a PWM wave…” —> “Firstly, a PWM wave at frequency … is provided…”
Also, definitions are lacking/missing.
It’s not clear what MOS tube ds is at all. It doesn’t seem to be mentioned elsewhere in the paper apart from Fig 3, when explaining V_{ds}. Please explain.
The parameters I_r and V_{ds} should be defined in the figure caption of Fig 3.
ASD is not defined anywhere. Please define. I believe it’s Amps per square decimeter. But I couldn’t find these words in your paper. I had to use Google.
Also first letters of new sentences are sometimes not capitalized.
Section 4 has problems. There are some parameters missing.
It’s not clear what the difference between figs 9a and 9b is. Please explain, preferably giving some indication in the caption.
The paragraph starting at line 305 is important for your paper. However, it is full of grammatical errors and looks hurriedly written. This is disappointing. Please consider rewriting this paragraph as several smaller ones. This can be done by either focusing on one figure at a time, or, focusing on one effect at a time. At least break the information into easier conceptual pieces. Perhaps make overarching comments at the beginning of this piece saying what you find. One approach to this is to label a new section at the beginning of this paragraph “Discussion,” because this is where you describe the results of your work.
Some of the errors I’ve found with include those following:
Consistency: 16 ASD or 16ASD? Check all the way through the paper.
I don’t understand what is intended by “resulting in a larger of the plated particle diameter 319 and the rate of change of hole wall thickness…” do you mean “resulting in an increase of the plated particle diameter 319 and the rate of change of hole wall thickness.”
“the size of the plated 322 particle diameter decrease and then increase,..” —> “the size of the plated 322 particle diameter decreases and then increases,..”
Also, where you say “are increased, This…” are you starting a new sentence here?
The conclusion is unsatisfying. You should at least mention the kinds of trends and what the optimum regions in the discussion immediately prior entail. You say that the LLC system works faster than a typical DC one. You don’t even say how much faster! Without giving any quantitative indicators, why would an industry person who might want to look at your system dig further? Please be more quantitative in your conclusion.
Author Response
Thank you for the affirmation of our work and the suggestions for revision. Now I will give you a reply to your suggestion.
Q1. There are many English grammar errors.
A1. Thank you for your suggestion. I have corrected all the English grammar errors you pointed out.
Q2. Definitions are lacking/missing.
A2. Thank you for your suggestion. I have added some definitions, including ASD, I_r and V_{ds}.
Q3. Format problem.
A3. first letters of new sentences are capitalized, and “16ASD->16 ASD”.
Q4. It’s not clear what the difference between figs 9a and 9b is. Please explain, preferably giving some indication in the caption.
A4. Figs 9a is cross-sectional view of via-holes orifice under bidirectional pulse electroplating, Figs 9b is cross-sectional view of via-holes orifice under DC electroplating. This is explained in the text and the caption.
Q5. The paragraph starting at line is important for your paper. However, it is full of grammatical errors and looks hurriedly written. This is disappointing. Please consider rewriting this paragraph as several smaller ones. This can be done by either focusing on one figure at a time, or, focusing on one effect at a time. At least break the information into easier conceptual pieces. Perhaps make overarching comments at the beginning of this piece saying what you find. One approach to this is to label a new section at the beginning of this paragraph “Discussion,” because this is where you describe the results of your work.
A5. With reference to your suggestion, I have revised this section into eight paragraphs. The first four paragraphs introduce 15 samples, the fifth paragraph describes the general situation of Figure 10 and Figure 11, and the last three paragraphs focus on the impact of frequency, current density and the duty cycle ratio of forward and reverse pulses on copper plating
Q6. The conclusion is unsatisfying. You should at least mention the kinds of trends and what the optimum regions in the discussion immediately prior entail. You say that the LLC system works faster than a typical DC one. You don’t even say how much faster! Without giving any quantitative indicators, why would an industry person who might want to look at your system dig further? Please be more quantitative in your conclusion.
A6. I have thought about this problem, but it is not so easy. I don't know how to describe "how fast" and "quantitative indicators" properly. I use pulse power supply and DC power supply with the same current amplitude for electroplating, and compare the coating thickness within the same time to obtain the rate, but I think this is unfair and meaningless, because in this case, the coating effect of the two is very different. In fact, I have described some "quantitative indicators" in the article, for example, “However, compared with pulse plating, the plating thickness is less than one quarter of pulse plating”, ”the plating thickness of the via-holes from outside to inside is uniform, all 25μm” and “the surface flatness decreases, and the thickness of the plating from the outside to the inside of the via-holes gradually decreases from 30μm to 15μm”. the comparative rate of pulse electroplating and DC electroplating with different current amplitude but similar electroplating effect., this is the most appropriate way I have thought of. This is a topic worthy of discussion, and I have been studying this aspect, hoping to further quantify these indicators.
Reviewer 2 Report
Authors have presented an article entitled “Development of Bidirectional Pulsed Power Supply and Its Effect on Copper Plating Effect of Printed Circuit Board Via-Filling” Though the manuscript is well written and organized but there is scope for further improving the quality of the draft before considering for publication.
Few minor comments are listed below:
1. Author should use the full name of the abbreviation (like DSP- Digital Signal Processor, etc.) when it is used for the first time in any section.
2. As author mentioned that the electronic products are becoming lighter, thinner, and smaller, developing towards multi-function, high density, and high speed using resin or others. Author should send the performance of other articles to make the contrast: Composites Part A: Applied Science and Manufacturing, 107427 (2023); Fibers and Polymers 19 (5), 1064-1073 (2018).
3. In Section 3.3 last paragraph, I0 may be written as I0 (subscript), same for f0.
4. In Figure 6, the x and y axes are confusing to understand. Please rectify it by properly mentioning both axes. make it bold in a font (such as resonant cavity current and output pulse current).
5. Authors nicely designed the results and discussion section, they should check the typos and grammatical mistakes in the revised version.
Author Response
Thank you for the affirmation of our work and the suggestions for revision. Now I will give you a reply to your suggestion.
Q1. Author should use the full name of the abbreviation (like DSP - Digital Signal Processor, etc.) when it is used for the first time in any section.
A1. Thank you for your suggestion. I have added the full name of the abbreviation.
Q2. As author mentioned that the electronic products are becoming lighter, thinner, and smaller, developing towards multi-function, high density, and high speed using resin or others. Author should send the performance of other articles to make the contrast: Composites Part A: Applied Science and Manufacturing, 107427 (2023); Fibers and Polymers 19 (5), 1064-1073 (2018).
A2. This paper mainly studies the copper plating of through-hole of PCB, but has not yet studied the material aspect. It will be studied in this direction sometime in the future.
Q3. In Section 3.3 last paragraph, I0 may be written as I0 (subscript), same for f0.
A3. Corrected
Q4. In Figure 6, the x and y axes are confusing to understand. Please rectify it by properly mentioning both axes. make it bold in a font (such as resonant cavity current and output pulse current).
A4.The x-axis represents time and the y-axis represents current amplitude. Time and current symbols have been added to the figure and are bold. The mark above the waveform is resonant cavity current and output pulse current.
Q5. Authors nicely designed the results and discussion section, they should check the typos and grammatical mistakes in the revised version.
A5. checked. Among them, the word "hole" has been reported incorrectly. Don't worry. I have verified it. It is no problem.
Reviewer 3 Report
The authors shown the analysis of the effect of bidirectional pulse power supply on via-filling copper plating, demonstrating the effect of pulse electrical parameters on the electroplating, even in comparison with DC Plating.
The paper is well written and the results are clearly explained, only a minor revision on English language can be suggested
Author Response
Thank you for the affirmation of our work and the suggestions for revision. I have improved the article.
Reviewer 4 Report
Please find the comments, attached.

Author Response
Thank you for the affirmation of our work and the suggestions for revision. Now I will give you a reply to your suggestion.
Q1. The most important research results should be developed in e.g. two sentences. This part of the data is too short in the Abstract.
A1. Many experimental conclusions are written in the abstract, and only one sentence of specific data is written, because this data only accounts for a part of the whole experimental conclusion.
Q2. Figure 1 should be improved. The arrows are unsightly.
A2. I can see what you mean. This arrow is designed specially. I think this arbitrary line is more beautiful.
Q3. Please standardize the ending of the text just before the mathematical formula. Sometimes there is a dot and sometimes there is no sign. I suggest putting a colon everywhere.
A3. Thank you for your suggestion. I've put a colon everywhere.
Q4. Figure 6 should be labeled as a part "a" and part "b" for better readability. In addition,
a description of the X axis should appear on the first part of the chart.
A4. Thank you for your suggestion. I think it is necessary to put these two figures together so as to highlight the function of PWM wave turning off during pulse turning off. Your suggestion is also very reasonable. After comprehensive consideration, the x-axis and y-axis labels are added and bold.
Q5. There is no data analysis in connection with works on similar topics in the article. There should be a few sentences referring to the comparison of the obtained results. For example, in the conclusion.
A5. There are researches on pulse electroplating power supply, electroplating technology and the influence of electrical parameters on electroplating, but these are not integrated. Moreover, the final electroplating effect will vary with different solution systems, so in order to ensure the preciseness, it is not compared with the conclusions in similar articles.
Reviewer 5 Report
The authors have presented "Development of Bidirectional Pulsed Power Supply and Its Effect on Copper Plating Effect of Printed Circuit Board Via Filling.
The paper and well written and nicely explained. However, I have some suggestions to improve the paper.
1. In the Abstract define or write DSP in full.
2. The language needs a little improvement.
3. How many semiconductors are used for the application.
4. What is the efficiency of the overall system.
5. Why isolated topologies are suitable for such applications.
6. Since the application is low voltage high current, why high voltage low current (STP18N65M2 type) is used?
7. what would be the impact of the GaN device which has excellent characteristics for such an application?
8. How to design a proper controller for such an application. The author can add a section if someone is interested in implementing the prototype described by the author. They can look at the reference for the controller design. I am suggesting some of them.
a) Iqbal, Mohammad Tauquir, Ali I. Maswood, Hossein Dehghani Tafti, Mohd Tariq, and Zhong Bingchen. "Explicit discrete modelling of bidirectional dual active bridge dc–dc converter using multi‐time scale mixed system model." IET Power Electronics 13, no. 18 (2020): 4252-4260.
b) V. Vorperian, “Approximate Small-Signal Analysis of the Series and the Parallel Resonant Converters,” Power Electronics, IEEE Transactions on, Volume: 4, Issue: 1 , Jan. 1989
Author Response
Thank you for the affirmation of our work and the suggestions for revision. Now I will give you a reply to your suggestion.
Q1. In the Abstract define or write DSP in full.
A1. Thank you for your suggestion. I have added the full name of the abbreviation.
Q2. The language needs a little improvement.
A2. We apologize for the poor language of our manuscript. We worked on the manuscript for a long time and the repeated addition and removal of sentences and sections obviously led to poor readability. We have now worked on both language and readability. We really hope that the flow and language level have been substantially improved.
Q3. How many semiconductors are used for the application.
A3.16. Full-bridge MOS (4) and rectifier diode (2) in forward LLC resonant circuit; Full-bridge MOS (4)and rectifier diode (2) in reverse LLC resonant circuit; Lead bridge arm MOS (2) of Bidirectional Pulse Generator; lag bridge arm MOS (2) of Bidirectional Pulse Generator. So, a total of 6 kinds, 16 semiconductors.
Q4. What is the efficiency of the overall system.
A4. the efficiency of the overall system can reach up to 93%. Thank you for your suggestion. This part has been supplemented in the article.
Q5. Why isolated topologies are suitable for such applications.
A5. Firstly, Security. Input voltage is generally 220V AC or 380V AC, Using isolated topology can greatly improve security. Secondly, Easy to realize step-down conversion. The output is about 10V low voltage, input voltage is generally 220V AC or 380V AC, the extent of this change is great. Using isolated topology can be realized easily, and can ensure high efficiency. Finally, it has strong anti-interference ability.
Q6. Since the application is low voltage high current, why high voltage low current (STP18N65M2 type) is used?
A6. Indeed, the output of the pulse power supply is low voltage and high current, but the input voltage of LLC is 310v, so the design of the power switch tube of LLC selects a SIHA25N60EFL type MOSFET with a maximum current of 25A and a breakdown voltage of 650V.
Q7. what would be the impact of the GaN device which has excellent characteristics for such an application?
A7. First and foremost, the GaN will improve the output efficiency greatly due to its low conduction internal resistance. Secondly, because of its larger bandgap width and smaller parasitic parameters, it can work at higher frequency and has higher reliability.
Q8. How to design a proper controller for such an application. The author can add a section if someone is interested in implementing the prototype described by the author. They can look at the reference for the controller design. I am suggesting some of them.
A8. Thank you for your suggestion. I carefully read the literature you recommended. I chose this controller based on actual demand and cost. But I didn't explain how to choose it in detail in the article, because I considered whether this part of the article would be wordy and pointless. Now, based on your suggestions, the article explains why should use this controller.